# Long-Term Effects of Air Pollutants on Mortality Risk in Patients with End-Stage Renal Disease

**DOI:** 10.3390/ijerph17020546

**Published:** 2020-01-15

**Authors:** Jiyun Jung, Jae Yoon Park, Yong Chul Kim, Hyewon Lee, Ejin Kim, Yong-Lim Kim, Yon Su Kim, Jung Pyo Lee, Ho Kim

**Affiliations:** 1Department of Biostatistics and Epidemiology, School of Public Health, Seoul National University, Seoul 08826, Korea; bestjudy@hanmail.net (J.J.);; 2Department of Internal Medicine, Dongguk University Ilsan Hospital, Gyeonggi-do 10326, Korea; 3Department of Internal Medicine, Dongguk University College of Medicine, Gyeongsangbuk-do 38066, Korea; 4Department of Internal Medicine, Seoul National University College of Medicine, Seoul 03080, Korea; imyongkim@gmail.com (Y.C.K.); yonsukim@snu.ac.kr (Y.S.K.); 5Department of Neuropsychiatry, Seoul National University Bundang Hospital, Gyeonggi-do 13620, Korea; woniggo@gmail.com; 6Institute of Health and Environment, Seoul National University, Seoul 08826, Korea; 7Department of Internal Medicine, Kyungpook National University College of Medicine, Daegu 41566, Korea; ylkim@knu.ac.kr; 8Department of Internal Medicine, Seoul National University Boramae Medical Center, Seoul 07061, Korea

**Keywords:** particulate matter, nitrogen dioxide, sulfur dioxide, ESRD, mortality

## Abstract

Long-term exposure to air pollutants significantly increases the morbidity and mortality associated with various diseases. However, little is known about the relationship between air pollutants and end-stage renal disease (ESRD)-related mortality. A total of 5041 patients who started dialysis between 2008 and 2015 were prospectively enrolled in the Clinical Research Center for End-Stage Renal Disease (CRC-ESRD) cohort study. We assigned a daily mean concentration of air pollutants (PM_10_, NO_2_, and SO_2_) to each participant. Time-varying Cox models were used to investigate the relationship between air pollutants and mortality in ESRD patients. During the follow-up period (mean 4.18 years), 1475 deaths occurred among 5041 participants. We found a significant long-term relationship between mortality risk and PM_10_ (HR 1.33, CI 1.13–1.58), NO_2_ (HR 1.46, CI 1.10–1.95), and SO_2_ (HR 1.07, CI 1.03–1.11). Elderly patients and patients who lived in metropolitan areas had an increased risk associated with PM_10_. Elderly patients also had increased risks associated NO_2_ and SO_2_. Long-term exposure to air pollutants had negative effects on mortality in ESRD patients. These effects were prominent in elderly patients who lived in metropolitan areas, suggesting that ambient air pollution, in addition to traditional risk factors, is important for the survival of these patients.

## 1. Introduction

As industrialization progresses, the impact of air pollution on health is becoming a global issue. According to the World Health Organization (WHO)’s recent statement, seven million premature deaths related to air pollution occur annually, particularly in the regions of southeastern Asia and the western Pacific [1]. Exposure to ambient air pollution contributes to the progression of cardiovascular (CV) diseases and respiratory diseases, such as chronic obstructive pulmonary disease and type 2 diabetes mellitus (DM), malignancies, and even autoimmune diseases [2,3,4,5,6].

The prevalence of chronic kidney disease (CKD) is increasing, and it has become one of the most important health problems worldwide. Individuals with CKD have an eight- to 10-fold higher risk of CV-related mortality than those without renal dysfunction [7,8]. DM, hypertension, dyslipidemia, smoking, old age, and male sex are well-established traditional risk factors for CKD [9,10]. However, recent reports have shown that air pollution, as well as traditional risk factors, play an important role in renal disease [11,12,13,14,15,16,17]. 

Recent epidemiologic reports emphasized that aerodynamic particulate matter less than 10 μm (PM_10_) significantly increases the morbidity and mortality of various diseases, especially CV and pulmonary diseases [2,3,18,19]. Biologically, particulate matter (PM) was reported to contribute to the exacerbation of chronic diseases by promoting inflammatory or carcinogenic responses, and evidence suggests that the mechanism is mediated by both organic and inorganic components of PM [20,21]. 

Although evidence from several experimental and clinical studies suggests that exposure to relatively high amounts of air pollution adversely affects kidney function [11,12,13,14], few studies have reported the association between mortality in CKD patients and PM. In addition, few researchers have found an association between gaseous compounds, such as nitrogen dioxide (NO_2_) and sulfur dioxide (SO_2_), and kidney disease, although epidemiological studies have demonstrated the effect of long-term exposure to gaseous matter on general health [22,23,24]. Therefore, we investigated the impact of air pollutants on mortality in patients with ESRD in a nationwide, multicenter, prospective cohort in Korea.

## 2. Materials and Methods 

### 2.1. Study Population

The Clinical Research Center for End-Stage Renal Disease (CRC-ESRD) cohort (Daegu, Republic of Korea) is a prospective cohort that has been continuously registered since July 2008 and includes those who have undergone dialysis previously. The number of hospitals contributing to the cohort was 31, which was advantageous because the characteristics of the majority of ESRD patients was able to be identified. Patients who started dialysis between 2008 and 2015 and those who started dialysis prior to cohort registration were included. The data from the CRC-ESRD cohort included age, sex, body mass index (BMI), hemoglobin, Charlson comorbidity index (CCI), working status, marital status, education, social and family support, and insurance type. Social and family support were measured as percentages (%) of the degrees to which the patient was assisted by their family and society. Zero percent and 100% represented independent and full-support on society/family, respectively. Insurance was divided into five categories: (1) Medical protection (type 1): Those who received the National Basic Livelihood Security; (2) Medical protection (type 2): Those who received medical protection but not type 1; (3) Health insurance, working poor: Those in the potential low-income strata; (4) Health insurance, rare and incurable disease: Those with rare intractable diseases designated by country; and (5) Health insurance, general: Those who were normal subscribers of National Health Insurance. Working and marital status referred to the status at the time of cohort entry.

### 2.2. Data Collection

Air pollution data were collected from the National Institute of Environmental Research database (Incheon, Republic of Korea). This database provides the concentrations of PM_10_, NO_2_, and SO_2_ from 274 stations on an hourly basis through the national Urban Atmospheric Monitoring Network. The whole country is classified into 17 provincial-level regions, including major metropolitan cities (Seoul, Busan, Incheon, Daegu, Gwangju, Daejeon, and Ulsan). We assigned daily mean concentrations of air pollutants to the 17 provincial-level regions (si-do) according to the locations of monitoring stations. The CRC-ESRD cohort data did not include the residential addresses of participants, but did include the province-level addresses of the hospitals in which the patients were treated. Therefore, exposures were assigned to the patients based on the province of the hospital where they received dialysis. Due to the characteristics of dialysis, the distance between the dialysis center and patient residence was likely not far because dialysis needs to be performed regularly [25], in general, three times a week. In addition, the following province-level variables were obtained from the Korean Statistical Information Service (Daejeon, Republic of Korea) (1) Population density: Population divided by land area (km^2^); (2) Economically active population: Mean population of economically active population aged over 15 years, including the number of employed and unemployed people per thousand people; and (3) Number of medical institutions: Mean number of institutions, including hospitals, clinics, public health agencies, and pharmacies. All province-level characteristics were matched to the enrollment year of the cohort.

### 2.3. Statistical Analysis

Time-varying Cox hazard models were used to investigate the relationship between mortality in ESRD patients who received dialysis and long-term air pollutants. The fully adjusted model is described as follows:(1)$$\lambda (t|Z\left(t\right))=\lambda_{0}\left(t\right)\exp\left(\beta^{\prime }x+\gamma^{\prime }Xg\left(t\right)\right)$$
(2)$$Z\left(t\right)=\left[x_{1},x_{2},\ldots ,x_{14},X_{1}g\left(t\right)\right]$$
where *λ*_0_
*(t)* is the baseline hazard function and *β*′ and *γ*′ are the coefficients of time-independent and time-dependent covariates, respectively. The models were adjusted for potential confounding factors, including sex (male, female); age (continuous); smoking status (never, current, former); hemoglobin, BMI, and CCI (continuous); working status (unemployed, retired, employed); marital status (single, married); education (uneducated, elementary, middle, and high school; university/college; graduate school); insurance (medical protection (type 1 or 2), health insurance (working poor, rare and incurable disease, general)); and the duration of therapy, population density, economically active population, and the number of medical institutions (continuous). The enrollment year of ESRD patients was additionally adjusted since the entry dates of the patients were different. In addition, *X*_1_
*g(t)* was used to describe the annual average exposure to air pollutants from one year to seven years before the date of cohort enrollment. We used exposure as a time-varying variable by updating each year. For example, one participant was enrolled in the cohort in 2008 and censored in 2011. To estimate the effect of six years of exposure on mortality, the mean concentrations of the 2002–2008, 2003–2009, 2004–2010, and 2005–2011 exposures were used. Long-term impacts for up to seven years could be identified, since the National Institute of Environmental Research in Korea has provided the daily concentration of air pollutants across the country since 2001. Person-years of follow-up time were calculated from the start of follow-up (29 August 2008) until the end of follow-up (31 December 2015), censoring at death or the end of follow-up. A regression spline with two degrees of freedom was used to identify nonlinearity between mortality and air pollutants in the time-varying Cox proportional hazard model. We selected the exposure period that yielded the smallest Akaike information criterion (AIC) to conduct subgroup analyses.

We stratified the fully adjusted model by age (<65, ≥65), sex (male, female), social and family support (<50%, ≥50%), and habitation in a metropolitan area (no, yes). Metropolitan cities included seven major cities with populations of more than one million in Korea. The statistical significance of the subgroup differences was confirmed by the *p*-value for interaction. The results are presented as hazard ratios (HRs) and 95% confidence intervals (CIs) per interquartile range (IQR) increment of each air pollutant. All statistical analyses were performed with R version 3.5.1 (R Foundation for Statistical Computing, Vienna, Austria).

### 2.4. Sensitivity Analysis

We constructed a time-independent baseline Cox model to identify the effect of PM_10_, NO_2_, and SO_2_ on mortality in ESRD patients. The effects of exposure to air pollutants from one year to seven years before the enrollment date of the cohort could be identified in the baseline model, although the time-varying model in which air pollutants were updated every year reflected the concentration of air pollutants over time. The trend *p*-value represents the linear trend of association considering the median value of each quartile as a continuous variable. In addition, we constructed a two-pollutant model with adjustments for other pollutants to identify the robustness of the association.

### 2.5. Validation Cohort

We used a validation cohort from Seoul National University Hospital (SNUH) from January 2001 to December 2017 to confirm the long-term association between mortality in ESRD patients and air pollutants (*n* = 8941). We excluded patients who were enrolled in the cohort before 2008 to identify the seven-year long-term effect of air pollutants on mortality. Overall, 5910 participants were analyzed with a time-varying Cox model adjusted for seven-year air pollutants, sex, age, BMI, hemoglobin, alcohol consumption, smoking, population density, economically active population, the number of medical institutions, and the year of cohort enrollment. We used the average of air pollutants measured at monitoring stations within 3 km, 5 km, and 10 km of the residence since the SNUH cohort data provided the residence of the participants.

### 2.6. Ethical Aspects

The study protocol complied with the Declaration of Helsinki and received full approval from the institutional review board (IRB) of Seoul National University Hospital (H-1405-060-579). We obtained informed consent from the participants prior to enrollment in the present study.

## 3. Results

### 3.1. Baseline Characteristics of the Subjects and Annual Trends of Air Pollutants 

Table 1 shows the characteristics of the 5041 participants in the CRC-ESRD cohort. During the mean 4.18-year follow-up period, 1475 deaths (29.2%) occurred. The mean age of the ESRD patients was 60.48 years and 59% were male. Most of the subjects were nonsmokers or former smokers (90%) and 73% were unemployed. Nearly half of the participants responded that they received more than 50% support from their family and society. The concentration of air pollutants during 2001–2015 showed a tendency to decrease slightly over the study period (Figure 1). The means and standard deviations of the concentrations of PM_10_, NO_2_, and SO_2_ measured seven years before the enrollment date of the cohort were 55.52 ± 5.28 μg/m^3^, 28.13 ± 6.89 ppb, and 5.60 ± 0.88 ppb, respectively. The IQR increments of seven-year exposure to PM_10_, NO_2_, and SO_2_ were 8.14 μg/m^3^, 12.77 ppb, and 0.38 ppb, respectively. The average concentrations of air pollutants slightly increased as the duration of exposure increased from one year to seven years (Appendix A). 

### 3.2. Association between Air Pollutants and Mortality of ESRD Patients in the CRC-ESRD Cohort

Figure 2 shows the association between mortality and the IQR increase in air pollutants. In the time-varying Cox model, the HRs per 7.54 μg/m^3^ increment in PM_10_ for one year (HR 1.10 CI 0.97–1.24), 6.65 μg/m^3^ increment for two years (HR 1.09, CI 0.97–1.24), and 7.6 μg/m^3^ increment for three years of exposure (HR 0.15, CI 0.99–1.34) were not statistically significant. However, long-term PM_10_ exposure, from four years to seven years, increased the mortality risk in ESRD patients from 15% to 33%. We estimated a similar risk for long-term exposure to NO_2_ and SO_2_. Seven years of exposure to NO_2_ (HR 1.45, CI 1.09–1.94) further increased the HR compared with one year of exposure (HR 1.33, CI 1.05–1.69). SO_2_ exposure was significantly associated with an increased mortality risk in ESRD patients when they were exposed to a 0.5-ppb increase for one year (HR 1.07, CI 1.03, 1.11) and a 0.38-ppb increase for seven years (HR 1.04, CI 1.01–1.07) before cohort enrollment. We selected an exposure period of seven years for PM_10_ and NO_2_ and one year for SO_2_ according to the AIC (Appendix A). In addition, we investigated the association between mortality and air pollutants based on a natural spline with two degrees of freedom in a time-varying Cox model using seven-year exposure for PM_10_ and NO_2_ and one-year exposure for SO_2_. A significant linear association was confirmed (*p*-values for PM_10_ 0.0010, NO_2_ 0.0174, and SO_2_ 0.0003), although the effect of PM_10_ and NO_2_ on mortality of ESRD patients showed a nonlinear tendency with decreased mortality at high concentrations (Appendix A). 

### 3.3. Stratified Analyses Considering Potential Confounders

Table 2 shows the HRs for IQR increases in average seven-year exposure to PM_10_ and NO_2_ and one-year exposure to SO_2_ in age-, sex-, family and social support-, and metropolitan-specific subgroups. We found that age ha a significant effect on the association between air pollutants (PM_10_: HR 1.34 (1.13, 1.59); NO_2_: HR 1.52 (1.13–2.03); and SO_2_: HR 1.07 (1.04–1.11)) and mortality in ESRD patients (*p*-value for interaction 0.02–0.03). The effect of PM_10_ (HR 1.48, CI 1.19–1.83) and SO_2_ (HR 1.08, CI 1.03–1.14) on mortality was greater in females than males, but the differences were not significant (Pinteract = 0.09–0.43). However, males had a higher risk of mortality after NO_2_ exposure than females (HR 1.49, CI 1.11–2.01). Increased risks were observed in those who received ≥50% social and family support, but the differences were not statistically significant, except for NO_2_. In addition, the participants who lived in metropolitan areas had increased mortality risks associated with air pollution, but the differences were not significant, except for PM_10_.

### 3.4. Sensitivity Analysis 

Appendix A shows the hazard ratio of mortality for each air pollutant for each exposure period in the baseline Cox model. The years of exposure, from one to seven years, were separate exposure variables. Similar results in the time-varying Cox model were identified, suggesting a significant linear association between air pollutants and mortality, except for seven-year exposure to NO_2_ and six-year exposure to SO_2_. The highest mortality risks associated with PM_10_, NO_2_, and SO_2_ were found for six years (HR 1.26, CI 1.07–1.47), seven years (HR 1.32, CI 1.00–1.75), and two years (HR 1.08, CI 1.03–1.13) of exposure, respectively. In addition, two-pollutant time-varying Cox models were constructed by including a second pollutant. Significant correlations were found between PM_10_ and SO_2_ (r = 0.65), PM_10_ and NO_2_ (r = 0.56), and SO_2_ and NO_2_ (r = 0.74) (Appendix A). SO_2_ and NO_2_ were not included in the two-pollutant model due to their high correlation. Robust associations were confirmed, although the statistical significance was not high in the two-pollutant model (Appendix A).

### 3.5. Association between Air Pollutants and Mortality of ESRD Patients in the Validation Cohort 

We analyzed the data of the ESRD cohort from the Seoul National University Hospital between 2008 and 2017. A total of 5910 patients with ESRD were enrolled, and 58.7% were male. During the follow-up period of a mean of 3.57 years, the mortality rate of the validation cohort was 29.6%. Exposure was determined by calculating the average concentration of air pollutants measured by the monitoring stations within the buffer regions of 3 km (*n* = 4627), 5 km (*n* = 5287), and 10 km (*n* = 5535). The HRs of PM_10_, NO_2_, and SO_2_ measured at monitoring stations within 3 km of the residences of participants were 1.22 (1.01–1.46), 1.15 (1.02–1.30), and 1.20 (1.10–1.32), respectively, when the IQR unit was 8.13 μg/m^3^ for PM_10_, 4.73 ppb for NO_2_, and 0.93 ppb for SO_2_ (Figure 3).

## 4. Discussion

In the present study, we evaluated the long-term effects of air pollutants on the mortality of patients undergoing dialysis using data obtained from a nationwide, multicenter, prospective cohort of ESRD patients. We found a linear association between air pollutants and mortality, although there was a decreased risk at high NO_2_ concentrations, similar to other studies [26,27]. The results indicated that increased exposure to PM_10_ from one to seven years increased the mortality risk, and long-term exposure to SO_2_ and NO_2_ was a significant risk factor for ESRD morality regardless of the exposure period. The associations of these air pollutants remained robust in the baseline and two-pollutant models and were significant in the elderly patients.

Previously, epidemiological studies investigated the long-term effects of air pollutants on mortality. Hart, et al. [28] reported percent increases in all-cause mortality of 4.3%, 8.2%, and 6.9% for PM_10_ (6 μg/m^3^), NO_2_ (8 ppb), and SO_2_ (4 ppb) in 53,814 men from U.S. trucking companies during 1985–2000. In addition, several studies confirmed the association between cause-specific morality and air pollutants [29,30,31]. Recently, a large population-based study demonstrated the effects of air pollutants on kidney disease. Kim, et al. [32] suggested that various air pollutants were related to the risk of kidney dysfunction in Korea. The researchers reported that a 10-μg/m^3^ increase in PM_10_ and a 12-ppb increase in NO_2_ were associated with decreases of 0.46 and 0.85, respectively, in estimated glomerular filtration rates (eGFRs). Although several studies have reported the relationship between air pollutant exposure and the deterioration of renal function [12,13,14,32], few studies have explored the direct relationship between air pollution and mortality in CKD patients [15,17]. A study that prospectively observed 256 elderly hemodialysis patients for two years reported that living in the Taipei Basin, which has relatively high levels of air pollutants, such as carbon monoxide, PM_10_, and NO_2_, is associated with protein-energy wasting and inflammation, as well as two-year mortality [17]. This study group further researched the impact of air pollution on the clinical outcomes of peritoneal dialysis (PD) patients [15,16] They reported that high environmental PM_2.5_ exposure was associated with an increased risk of one-year PD-related infection [16] and that high NO_2_ exposure was associated with two-year mortality [15]. However, these studies enrolled a relatively small number of patients (maximum of 256 patients), and the observation periods were too short to evaluate the long-term effects of air pollution on clinical outcomes (maximum of two years).

Several hypotheses have been proposed to explain the biological effects of air pollutants on mortality. The mechanisms of inhaled particles causing renal function deterioration are similar to those proposed for CV diseases, such as inflammation and oxidative stress [12,14,33]. Evidence from laboratories has suggested that exposure to PM causes renal hemodynamic impairment and promotes oxidative stress, inflammation, and DNA damage in kidney tissue, which aggravates acute kidney injury and further progresses to chronic renal failure in murine models [34,35]. NO_2_, mainly derived from the industrial burning of fuels and transportation, and SO_2_ generated as industrial byproducts have been associated with respiratory symptoms [36,37]. However, air pollutants might affect remote organs, such as the kidneys, through the bloodstream [38,39]. 

In the subgroup analysis, we found that elderly patients had significantly greater risks associated with various air pollutants than participants <65 years of age. Our findings agree with previous evidence that older age groups are more vulnerable to air pollution than younger age groups [40,41]. In addition, the risks of sex-specific exposure differed by air pollutant, although the differences were not significant in our study. Several studies examining the relationship between air pollution and kidney disease have shown inconsistent sex-specific effects [12,42]. We found a distinct association between PM_10_ exposure and mortality in patients in metropolitan areas. Our findings suggest that residence locations in rural or urban areas are potential confounders of the long-term effects of air pollutants [43].

Our study had several limitations. First, in some areas, the measurements of air pollutants might be inaccurate because the number of stations at the province level varied by region. The number of monitoring stations was sufficient to cover the entire area of the metropolitan cities with high population densities. However, there were only seven monitoring stations in large provinces, such as Gang-won, which has a relatively small population. In addition, there could be ambient air pollution measurement errors by the monitoring stations in time-series studies. Second, the unit of province-level exposure was larger than the unit of district-level exposure. Our study might suffer from exposure errors between ambient air pollution and actual exposure. Third, exposure values were assigned based on the addresses of hospitals because information on actual residence was not provided for the CRC-ESRD cohort. However, it could be assumed that there would be a similar degree of exposure because the distances between the actual residence of the patients and the dialysis centers were within the same daily life radius. To account for the province-level exposure unit of the dialysis hospital, we used detailed exposure values measured within various buffer diameters (3 km, 5 km, and 10 km), established by drawing circles around the actual residences of patients in the validation cohort, independent of the CRC-ESRD cohort. We found that long-term exposure to air pollutants was significantly associated with increased ESRD mortality regardless of the buffer size. Fourth, other socioeconomic status factors, such as income and occupation, acted as confounding factors on a small scale. Fifth, our validation cohort was not entirely independent of the main cohort because one of the participating hospitals in the main cohort was SNUH. However, the two cohorts had some independent characteristics because of the different numbers of years tracked and the differences in participating hospitals. Last, it is impossible to estimate the hazard ratio separately for CV and non-CV disease because the number of fatalities caused by CV disease (*n* = 325) was not sufficient to estimate the coefficients.

## 5. Conclusions 

Long-term exposure to air pollutants had negative effects on mortality in ESRD patients. These effects were prominent in elderly patients who lived in metropolitan areas, meaning that ambient air pollution, in addition to traditional risk factors, was important to the survival of these patients. Further biological studies are needed to confirm the specific pathophysiology of PM_10_ consisting of various organic and inorganic molecules, as well as NO_2_ and SO_2_

## Figures and Tables

**Figure 1 ijerph-17-00546-f001:**
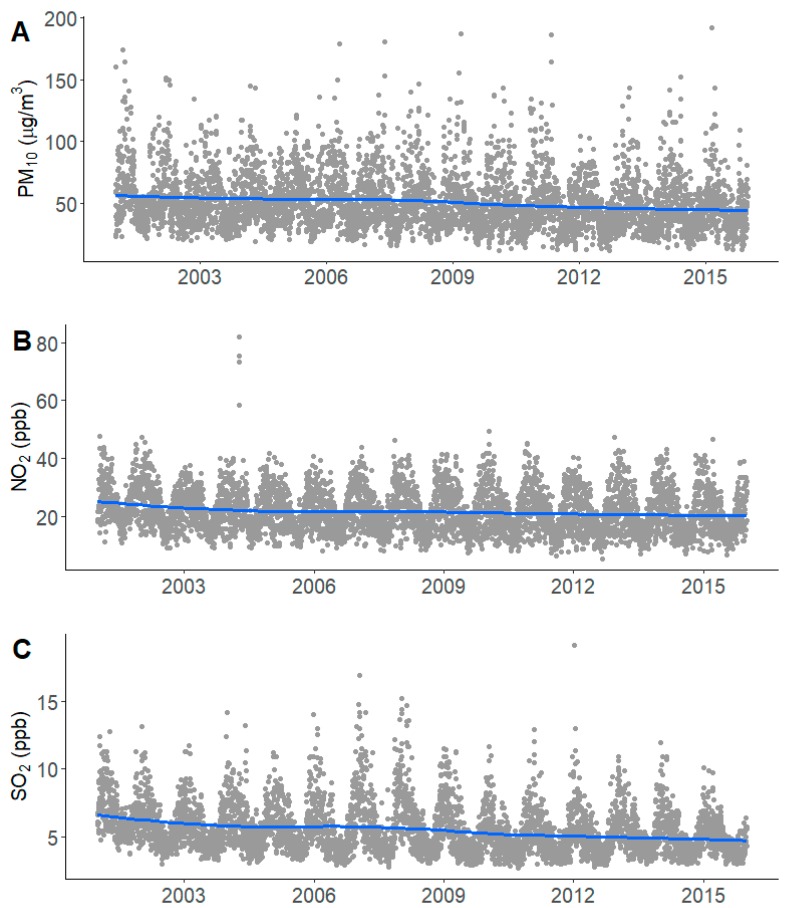
Distribution (grey dot) and smoothing trend (blue line) of mean PM_10_ (**A**), NO_2_ (**B**), and SO_2_ (**C**) concentration from 2001 to 2015 in Korea.

**Figure 2 ijerph-17-00546-f002:**
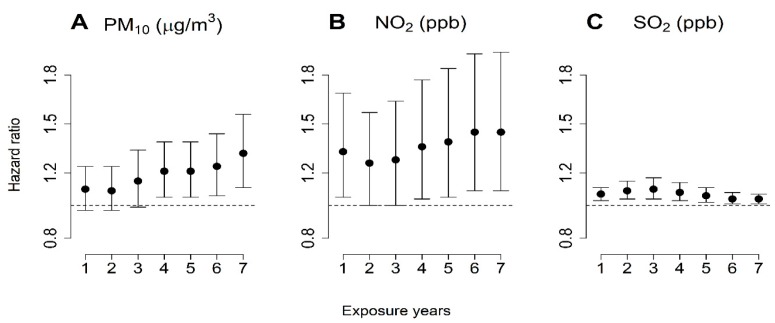
Hazard ratio and 95% confidence interval (CI) for end-stage renal disease (ESRD) mortality by duration of exposure to PM_10_ (**A**), NO_2_ (**B**), and SO_2_ (**C**) in a time-varying Cox model adjusted for sex, age, smoking status, hemoglobin, body mass index (BMI), Charlson comorbidity index (CCI), duration of therapy, working status, marital status, education, insurance, population density, and the number of medical institutions.

**Figure 3 ijerph-17-00546-f003:**
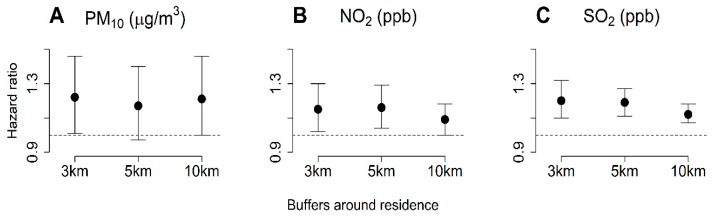
Hazard ratio and 95% CI per interquartile range (IQR) increase for air pollutants measured at monitoring stations within various buffer around the residence in validation cohort.

**Table 1 ijerph-17-00546-t001:** Descriptive characteristics of the Clinical Research Center (CRC) cohort participants at baseline.

Characteristics			Statistics
No. of cohort participants			*n* = 5041
No. of deaths			1475 (29.2%)
Person-years of follow-up			4.18 ± 1.77
Hemodialysis			5041 (100%)
Primary causes of ESRD			
	Hypertension		959 (19%)
	Primary glomerulonephritis (GN)		660 (13.1%)
	Diabetes		226 (4.5%)
	Cystic, hereditary, congenital disease		141 (2.8%)
	Secondary GN, vasculitis		65 (1.3%)
	Interstitial nephritis		53 (1%)
	Unknown		2821 (56%)
	Miscellaneous conditions		116 (2.3%)
Duration of therapy			0.84 ± 2.04
Individual level			
	Men		2963 (59%)
	Age		60.48 ± 13.52
	Hemoglobin (g/dL)		9.87 ± 1.67
	Smoking status	Never	2915 (60%)
		Current	491 (10%)
		Former	1463 (30%)
	CCI		5.09 ± 2.27
	Comorbidities	Cerebrovascular disease	677 (13%)
		Diabetes	545 (10%)
		Coronary artery disease	319 (6%)
		Cancer	314 (6%)
		Congestive heart failure	186 (3%)
	BMI		22.85 ± 3.38
	Working status	Unemployed	3576 (73%)
		Retired	286 (6%)
		Employed	1012 (21%)
	Marital status	Single	1148 (24%)
		Married	3562 (76%)
	Education	Uneducated	205 (4%)
		Elementary school	738 (16%)
		Middle school	732 (16%)
		High school	1781 (38%)
		University/college	1104 (23%)
		Graduate school	151 (3%)
	Social support	0%	986 (20%)
		<50%	1315 (27%)
		50–100%	2060 (42%)
		100%	579 (12%)
	Family support	0%	577 (12%)
		<50%	1136 (23%)
		50–100%	2402 (49%)
		100%	825 (17%)
	Insurance	Medical protection (1 type)	706 (14%)
		Medical protection (2 types)	48 (1%)
		Health insurance, working poor	144 (3%)
		Health insurance, rare & incurable disease	922 (19%)
		Health insurance, general	3131 (63%)
	Enrollment year	2008–2009	1514 (30%)
		2010	1371 (27%)
		2011	988 (20%)
		2012	646 (13%)
		2013	357 (7%)
		2014–2015	165 (3%)
Province level			
	Population density		6857.1 ± 6792.9
	Economically active population		3235.9 ± 2254.7
	Number of medical institutions (per 1000)		0.53 ± 0.02

**Table 2 ijerph-17-00546-t002:** The modification of association between interquartile range (IQR) increase of long-term exposure to air pollutants and mortality of end-stage renal disease (ESRD) patients by baseline characteristics among 5041 participants.

	PM_10_ (μg/m^3^) ^a^		NO_2_ (ppb) ^a^		SO_2_ (ppb) ^b^	
	HR (95% CI)	*P* _interact_	HR (95% CI)	*P* _interact_	HR (95% CI)	*P* _interact_
Age						
<65	1.30 (1.10, 1.54)		1.37 (1.02, 1.84)		1.05 (1.01, 1.09)	
≥65	1.34 (1.13, 1.59)	0.03	1.52 (1.13, 2.03)	0.02	1.07 (1.04, 1.11)	0.02
Sex						
Female	1.48 (1.19, 1.83)		1.37 (0.99, 1.89)		1.08 (1.03, 1.14)	
Male	1.25 (1.04, 1.49)	0.09	1.49 (1.11, 2.01)	0.43	1.06 (1.02, 1.10)	0.39
Family support						
<50%	1.30 (1.10, 1.55)		1.42 (1.06, 1.90)		1.06 (1.02, 1.10)	
≥50%	1.32 (1.12, 1.56)	0.22	1.48 (1.11, 1.98)	0.13	1.07 (1.03, 1.11)	0.26
Social support						
<50%	1.30 (1.09, 1.54)		1.42 (1.06, 1.90)		1.06 (1.02, 1.10)	
≥50%	1.32 (1.11, 1.56)	0.08	1.51 (1.12, 2.02)	0.04	1.07 (1.03, 1.11)	0.12
Metropolitan						
No	1.27 (1.06, 1.52)		1.29 (0.78, 2.14)		1.04 (0.99, 1.09)	
Yes	1.33 (1.12, 1.59)	<0.01	1.37 (0.97, 1.95)	0.58	1.06 (1.02, 1.10)	0.12

^a^ Seven-year average concentration before cohort enrollment; ^b^ one-year average concentration before cohort enrollment.

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
