# Peer review of "Long-Term Effects of Air Pollutants on Mortality Risk in Patients with End-Stage Renal Disease"

_ijerph, 2020, doi:10.3390/ijerph17020546_

Round 1
Reviewer 1 Report
Thank you for the interesting paper.
Maior points:
The definition of the exposures is unclear. It should be made much clearer if the numerical concentration of pullutants or the concentration split into quartiles is used. Moreover, it is unclear if the exposure years (e.g. in Table S3) are to be understood as separate exposure variables or as mean over the last (e.g. three) years.
The definition of the model is unclear. It seems like the exposure is measured before inclusion, but the risk time in Cox regression is after inclusion. The HR are reported as possible variyng nonlinearly with exposure level, but many places in the manuscript time-varying Cox regression is mentioned, without any clear explanation of, what is varying by time.
In the limitations section, you should discuss possible confounding by other factors, than those adjusted for, as e.g. other indicators of socioeconomic status clearly could confound small scale place of living (And hence air pollution) as well as health.
You are adjusting for some characteristics at inclusion that could be intermediary factors, as they themselves might be influenced by air pollution, e.g. comorbidities.
Figure 2 is missing.
It is not entirely clear if the validation cohort is disjoint from the main cohort.
Reviewer 2 Report
In this manuscript, authors demonstrated that long term exposure to the air pollutants such as PM10, nitrogen oxide (NO2), and sulfur dioxide (SO2) could be associated with increased risk of mortality in patients with end stage renal disease (ESRD), who were enrolled in Korean prospective cohort study. The subject of study seems to be interesting. However, there are some concerns in this study. The reviewer’s comments are described as follows.
1. In this manuscript, Figure 2 is missing. As this data is most important, it is impossible to appropriately review this manuscript.
2. As all subjects had been undergone dialysis therapy, dialysis modalities have to be included in Table 1. In addition, primary causes of ESRD and representative comorbidities such as hypertension, diabetes, cancers and cardiovascular diseases should be also added in Table 1.
3. According to the methods, authors adjusted mortality by sex, age, smoking status, hemoglobin, BMI, CCI, working status, marital status, education, insurance, population density, and the number of medical institutions. However, mortality in ESRD patients has to be also adjusted comorbidities as described above as well as duration of dialysis therapy.
4. In this study, the definition of “ESRD-related mortality” is unclear. As authors described in discussion, air pollutants can be associated with increased risk of cardiovascular diseases. However, sustained inflammation and DNA damage can lead to the development of cancers. Authors should clearly describe the definition of “ESRD-related mortality”. If possible, authors should separately analyze the mortality as cardiovascular and non-cardiovascular. If such analysis is impossible, authors should describe this issue as a limitation.
5. Authors should present data to explain whether the same main results as shown in Figure 2 were reproduced in the validation cohort.
6. Authors must describe the name of ethical committee and the approval number assigned by the ethical committee.
Round 2
Reviewer 1 Report
Thank you for your responses, they have improved the manuscript considerable. I have only one small follow-up suggestions:
Please add an explanation similar to your "Response 2" to the method section (or if not possible because of space constraints to the supplementary materials), as your explanation of time-varying exposure windows is important for the readers.
Author Response
Point 1: Thank you for your responses, they have improved the manuscript considerable. I have only one small follow-up suggestions:
Please add an explanation similar to your "Response 2" to the method section (or if not possible because of space constraints to the supplementary materials), as your explanation of time-varying exposure windows is important for the readers.
Response 1: Thank you for the great comments. We added the explanation in the manuscript as follows; “We used exposure as a time-varying variable by updating each year. For example, one participant was enrolled in the cohort in 2008 and censored in 2011. To estimate the effect of 6 years of exposure on mortality, the mean concentrations of the 2002-2008, 2003-2009, 2004-2010, and 2005-2011 exposures were used.” (page 3, line 118)

Reviewer 2 Report
Authors have satisfactorily addressed the reviewer's concerns. The paper can be accepted.
Author Response
Thank you for the great review.